# Detection of Gluten in Gluten-Free Foods of Plant Origin

**DOI:** 10.3390/foods11142011

**Published:** 2022-07-07

**Authors:** Jana Výrostková, Ivana Regecová, František Zigo, Slavomír Marcinčák, Ivona Kožárová, Mariana Kováčová, Daniela Bertová

**Affiliations:** 1Department of Food Hygiene Technology and Safety, University of Veterinary Medicine and Pharmacy in Košice, Komenského 73, 041 81 Košice, Slovakia; jana.vyrostkova@uvlf.sk (J.V.); slavomir.marcincak@uvlf.sk (S.M.); ivona.kozarova@uvlf.sk (I.K.); mariana.kovacova@uvlf.student.sk (M.K.); daniela.becova@uvlf.student.sk (D.B.); 2Department of Nutrition and Animal Husbandry, University of Veterinary Medicine and Pharmacy in Košice, Komenského 73, 041 81 Košice, Slovakia; frantisek.zigo@uvlf.sk

**Keywords:** DNA concentration, isolation, purity, gluten, PCR, plants foods

## Abstract

The work deals with the issue of standardization and more accurate methodology for the isolation of gluten DNA in gluten-free products of plant origin, which is more demanding due to the more complex structure of plant cells. Three isolation methods were compared, of which the combination of glass and zirconium beads, Proteinase K and a commercially produced isolation kit was confirmed to be the most effective procedure. The given isolation procedure was more effective in one-component gluten-free foods, where the concentration of the obtained DNA ranged from 80.4 ± 0.7 to 99.0 ± 0.0 ng/µL. The subsequent PCR reaction revealed the presence of gluten not only in guaranteed gluten-free products (40%), but also in naturally gluten-free foods (50%). These were mainly gluten-free sponge cakes, gluten-free biscuits “Cranberries”, cocoa powder, coffee “3in1”, and instant coffee.

## 1. Introduction

Gluten is a group of proteins found in cereal grains, of which wheat is the most consumed. The two main proteins found in gluten are gliadin and glutenin. The main role of these proteins is to create typical dough properties such as consistency, elasticity or ability to grow during baking. At the same time, they are necessary to maintain the texture, volume, satisfactory spread, durability and sensory quality of bakery products. Gliadin is responsible for most of the negative effects on the health of sensitive people [1,2].

Most people tolerate the presence of gluten in food, but there are currently several health problems caused by the gluten content of the diet. The most well-known diseases include celiac disease, gluten sensitivity, wheat allergy, and many others.

The most serious form of gluten intolerance is celiac disease. It is an autoimmune disorder that affects about 0.7–1% of the population. The most common symptoms of celiac disease are indigestion, bloating, diarrhea, constipation, headache, fatigue, rashes, depression, weight loss, and more. Ultimately, this disease damages the mucous membrane of the small intestine and can cause complete deficiency of nutrients in the body, anemia, severe indigestion, or an increased risk of many other diseases such as inflammation of the thyroid gland or type 1 diabetes mellitus. In some cases, the symptoms do not appear in the typical form, for example, fatigue or anemia are present, and for this reason it can be very difficult to diagnose celiac disease [3,4].

Celiacs tolerate small amounts of gluten, which are very individual. It is for this reason that people suffering from celiac disease do not rely only on 100% gluten-free foods, so they may have a reduced gluten content. The reason for using such foods is the technically and financially very demanding production of foods with zero gluten. The offer of gluten-free foods is dominated by foods with a low residual gluten content [5].

About 10% of the world’s population follows a strict gluten-free diet [6,7,8]. Gluten-free foods are those in which the total gluten level does not exceed 20 ppm [9]. There are specific regulations on gluten-free labeling worldwide. Most of them are based on the Codex Alimentarius standard 118-1979 and recommend good manufacturing practice to avoid cross-contamination with gluten in different countries. The European Union, the United States and Canada follow the limits of the GFF Code (20 ppm) [10,11]. Commission Implementing Regulation (EU) No. 828/2014 of 30 July 2014 [10] on the requirements for the provision of information to consumers on the absence or reduced content of gluten in food distinguishes between two basic indications: “gluten-free” and “very low gluten”. The term ‘gluten-free’ can only be used if the food, as sold to the final consumer, does not contain more than 20 mg/kg of gluten. The expression “very low gluten” may only be used if the food, which contains or consists of one or more ingredients made from wheat, rye, barley, oats or their cross-variants, specially processed to reduce gluten, does not contain more than 100 mg/kg of gluten in the food as sold to the final consumer. For people sensitive to gluten, the crossed ear symbol can ease food selection. This symbol may be of any color and may be supplemented by the words “gluten-free” in the relevant language. The symbol is not mandatory on the packaging, but is recognized worldwide and will help celiacs choose suitable foods [12]. In Argentina, a threshold of 10 ppm GFF is set [13]. In Australia and New Zealand, the legislation is stricter and states that in order for food to be considered “gluten-free”, it must not contain detectable gluten [14,15]. In Brazil, the legislation imposes an obligation, including a declaration of the presence or absence of gluten on the labeling of industrial products. However, it does not address the tolerable gluten limit [16].

Current methods for determining gluten do not allow reliable and repeatable measurement of gluten at levels below 20 ppm. There are several different analytical methods and technologies for testing food for allergens: LFT, ELISA, PCR. At present, the amount of gluten in foodstuffs in the Slovak Republic as well as in other EU countries is determined according to ISO 21415 and ELISA, where the detection limit is 4 ppm gluten [17,18].

The PCR method is also used for the detection of gluten in food, where the basis is to obtain high-quality pure DNA suitable for amplification. Fresh plant food products without any processing are suitable for many types of analytical or molecular analysis but, since most plant food samples are processed to some extent, DNA is usually altered and fragmented into small fragments [19].

In addition, DNA extraction from plant food products has several problems and limitations and requires special treatment [20]. Plant material goes through a number of processes that lead to DNA degradation. Thus, it may not lead to satisfactory PCR amplification, as the method requires intact targets. Therefore, PCR products should be shorter than 200 bp in order to enhance optimal results [21]. In addition, food products contain a wide range of substances, including carbohydrates, fats and chemicals, which often inhibit the PCR reaction, leading to false results (negative or positive) [22,23].

The offer of gluten-free products on the market is very diverse. Buying these products is costly and the price does not always match the quality. The assortment includes many mixes for baking bread, pastries and cakes, pasta, etc., but the offer of fresh pastries is relatively limited. Research has shown that the majority, i.e., 70% of celiacs, prepare bread at home from pre-prepared mixtures [24].

A safe and effective prevention for most patients with celiac disease is a gluten-free diet—a complete omission of gluten in food. At present, however, following such a diet is relatively expensive, inconvenient, and difficult to maintain. In most cases, gluten-free foods are processed in a common facility together with those that normally contain gluten, and thus cross-contamination may occur. Polymerase chain reaction (PCR) is used to detect trace amounts of gluten in naturally gluten-free foods. It is one of the basic molecular genetic methods and allows in vitro amplification of the target DNA sequence in a matter of hours in multiples of a billion [25].

Based on these facts, the development of a more efficient procedure for the isolation of gluten DNA and the subsequent detection of trace amounts of gluten in gluten-free foods of plant origin is approached.

## 2. Materials and Methods

Samples of the following foods were used to detect gluten in foods of plant origin that, due to the composition of the food and the process of production of the food, are not initially expected to contain gluten: gluten-free biscuits (Slovakia; less than 20 ppm gluten), corn crisps (Slovakia; less than 20 ppm gluten), powder vanilla pudding (Slovakia; less than 20 ppm gluten), gluten-free crackers by “Brusienky“ (Slovakia; less than 20 ppm gluten), “Crispins” corn sticks (Czech Republic; less than 20 ppm glutenu), “Promix UNI” gluten-free flour (Slovakia; less than 20 ppm gluten), “Felicia Bio” gluten-free pasta (Italy; less than 20 ppm gluten), instant coffee, “3in1” coffee (Czech Republic), cocoa powder (Slovakia) and corn starch (Slovakia). The food for analysis was purchased from a regular commercial network, with two samples of each type of food.

### 2.1. DNA Isolation

A sample of 20 mg was taken from each food examined. Subsequently, the DNA required for the detection of gluten by the PCR method was isolated from the sample thus taken. Isolating DNA from plant foods is very challenging due to the cell wall of the plant cells. However, for the detection of gluten by the PCR method, a sufficient amount of whole and pure DNA is required so that the DNA is not contaminated or disrupted during the isolation procedure. Taking into account the above aspects, we proposed three methods for isolating DNA from food of plant origin and determined which of them is the most suitable for further testing in terms of quality and quantity of isolated DNA. It was also determined which DNA isolation procedure is more suitable for samples of single-component and multi-component foods of plant origin. The concentration and purity of the DNA obtained by all three methods were detected using a BioSpec nanometer (SHIMADZU, Kyoto, Japan). Isolation procedures were repeated three times on individual samples for more objective evaluation of the results.

#### 2.1.1. The First Isolation Procedure

To a 1.5 mL microtube with a 20 mg of a sample, 200 μL of Triton-X, a nonionic surfactant used to isolate mainly bacterial DNA, was added. Subsequently, the sample was placed in a centrifuge and centrifuged at 12,000× *g* for 5 min. After centrifugation, the supernatant was removed and Triton-X was added to the sediment again in an amount of 300 µL. After vortexing, the diluted sediment microtube was heated in a dry bath at 95 °C for 10 min. After heating, the sample was cooled on ice for 5 min and then centrifuged again at 12,000× *g*/5 min (the modified method of Sharma et al. [26]). The resulting supernatant was a source of DNA template for further use in the PCR reaction.

#### 2.1.2. The Second Isolation Procedure

The 1.5 mL microtubes contained a sample of 20 mg of sterile zirconium and glass beads (1:1) in the volume of 0.2 mL, to better disrupt the cell wall. Proteinase K (Macherey Nagel, Dueren, Germany) was added to the microtubes thus prepared in an amount of 10 µL. The samples were incubated at 37 °C for 30 min. After incubation, the samples were vortexed vigorously for 1 min. Subsequently, 200 µL of Triton-X was added to the microtube. The sample thus prepared was centrifuged at 12,000× *g*/5 min. After centrifugation, the supernatant was removed and Triton-X was added to the sediment again in an amount of 300 µL. After vortexing, the diluted sediment microtubes were heated in a dry bath at 95 °C for 10 min. After heating, the sample was centrifuged again at 12,000× *g* for 5 min. The resulting supernatant was the source of DNA (the modified method of Sharma et al. [26]).

#### 2.1.3. The Third Isolation Procedure

Samples in a volume of 20 mg and sterile 0.2 mL zirconium and glass beads (1:1) were placed in 1.5 mL microtubes. Subsequently, Proteinase K (Macherey Nagel, Germany) was added in an amount of 10 µL. The samples were incubated at 37 °C for 30 min. After incubation, the sample was vortexed vigorously for 1 min (method developed in this study). After mixing, 800 µL of FG1 lysis solution, which is part of the commercial Fungal DNA Mini Kit (Omega Bio-Tek, Norcross, GA, USA), was added. The samples were incubated at 65 °C for 10 min. Subsequently, DNA was isolated according to the procedure of E.Z.N.A.^®^ Fungal DNA Mini Kit (Omega Bio-Tek, USA). The obtained supernatant was used as a source of DNA in PCR reactions.

### 2.2. Detection of Gluten Using PCR

#### PCR Amplification

The forward primer WBR11 (GGTAACTTCCAAATTCAGAGAAAC) together with the reverse primer WBR13 (TCTCTAATTTAGAATTAGAAGGAA) amplified a specific DNA fragment encoding gluten (200 bp). These primers were designed based on gluten sequences available from the GenBank-European Laboratory database and according by Olexová et al. [27].

The polymerase chain reaction was performed in a volume of 20 μL containing 1 ng to 10 ng DNA, 0.5 µL each primer (10 pmol/μL concentration), 4.0 μL HOT Firepol^®^ BlendMaster Mix (SolisBioDyne, Tartu, Estonia). Amplification was performed in a Techne thermocycler (London, UK) with an initial step of 95 °C for 12 min. The initial denaturation was followed by 30 cycles with the following steps: 95 °C/20 s; 52 °C/60 s and 72 °C for 2 min. After 30 cycles, a final extension of 72 °C was followed for 10 min. After amplification, the samples were cooled to 6 °C.

All PCR products were size fractionated on agarose gels (1.5%) and visualized using Nucleic Acid GelRed^®^ (Biotium Inc., Fremont, CA, USA). PCR products were visualized by UV transillumination using Mini Bis Pro^®^ (DNR Bio-Imaging Systems Ltd., Neve Yamin, Israel).

### 2.3. Sensitivity of Specific Detection Test

Sequences obtained from the studied food samples used in this work were submitted to the GenBank-EMBL database. DNA sequences obtained from amplified isolated DNA were homologously searched with sequences available in the GenBank-EMBL database using the BLAST program (NCBI software package, 2.13.0, Bethesda, MD, USA).

A DNA sample from semi-coarse wheat flour and dark wheat bread was used as a positive sample in both the isolation and PCR reaction.

### 2.4. Statistical Analysis

Numerical data presented in this study are expressed as the mean value ± standard deviation. Confidence interval was set at 95%. The statistical analysis was performed by One-way Analysis of Variance (ANOVA) and Tukey test for multiple comparison conducted with statistics software GraphPad Prism 8.3.0.538 (GraphPad SOFTWARE, San Diego, CA, USA).

## 3. Results

In our study, 22 samples of various types of plant foods from various manufacturers purchased from a regular commercial network were tested for gluten detection using the PCR method. Depending on their composition, these foods were either naturally gluten-free (corn crisps, instant coffee, “3in1” coffee, cocoa, corn starch) or declared as gluten-free by a manufacturer (gluten-free sponge cakes, vanilla pudding, gluten-free “Brusienky”, corn sticks, “Crispins”, gluten-free flour “Promix UNI”, gluten-free pasta Felicia Bio).

### 3.1. DNA Isolation from Food Samples Used as a Positive Control

The number of components that the product contains plays an important role in obtaining DNA from these products. This can lead to inhibition of DNA isolation. For this reason, the prepared isolation procedures were first tested on samples of a one-component (semi-coarse wheat flour) and a multi-component (dark wheat bread) food containing gluten. The yield of the isolation procedures was verified spectrophotometrically using nanodrop, and the usability and integrity of the isolated DNA was verified by a PCR reaction, which amplified a specific sequence encoding gluten.

As can be seen from Table 1, in a single-component reference sample (semi-coarse wheat flour), we were able to isolate DNA with higher average concentrations than the two-component samples (dark wheat bread) using all three methods. In terms of DNA purity, the isolated DNA was purer in the case of the two-component sample. The lower purity of DNA in single component food can be explained by its loose consistency, which caused an increase in sediment when handling the sample.

Figure 1 shows a comparison and verification of three isolation procedures for wheat semi-coarse flour and dark wheat bread samples by PCR method. Isolation procedure no. 1 was not efficient in terms of the DNA integrity of the isolation by PCR reaction. This may be due to low purity, insufficient DNA concentration and disruption of the integrity of the isolated DNA, which led to inhibition of amplification of the specific gluten sequence. As can be seen from Figure 1, isolation procedure no. 2 was able to isolate DNA from samples of either single- or multi-component food in sufficient concentration. However, the concentration of isolated DNA was lower than in procedure no. 3, which also can be seen in the radiation intensity of individual fragments in the agarose gel after exposure to UV radiation.

To determine the effectiveness of the isolation procedures, we also isolated DNA from a sample that is expected to have no or very low amounts of gluten. Isolation procedure no. 3 was found to be more effective in isolating DNA from the above gluten-free food sample. At the same time, procedure no. 3 appeared to be the most effective, whether in terms of concentration, purity or integrity of the DNA obtained in all samples initially examined.

### 3.2. DNA Isolation from Single-Component Food Samples

The individual samples were divided into two groups according to whether they contained only one component (single-component) or multiple components (multi-component) such as additives, additives and so on. The one-component samples were: cocoa powder, corn starch, vanilla pudding powder, instant coffee and gluten-free flour “Promix UNI”.

Since the number of components that the products contain is also very important in obtaining DNA from the products, the comparison shows that we managed to isolate a higher amount of DNA from one-component samples, using all three methods. This fact can be caused by the composition of the samples themselves or by the preparation of the samples before the isolation itself, as these are loose products and thus easier to process and break down.

Table 2 shows that the most advantageous isolation procedure for this group of products was no. 3, which allowed us to isolate the highest average DNA concentrations, and the obtained concentrations were also the purest. Isolation procedure no. 1, on the other hand, obtained the lowest DNA concentrations, and the DNA purity obtained showed low values. This procedure was the least time consuming, but ultimately ineffective. As for the insulation procedure no. 2, the use of Proteinase K and zirconium and glass beads proved to be relatively effective, but compared to process no. 3 was still insufficient to obtain the highest possible concentrations of DNA.

### 3.3. DNA Isolation from Multi-Component Food Samples

In the group of multi-component foods we have included: gluten-free sponge cakes, corn crisps, gluten-free “Brusienky”, corn sticks “Crispins”, gluten-free pasta “Felicia Bio”, and coffee “3in1”.

Efficiency of isolation procedure no. 3 was confirmed, as in the case of single-component foods, and also in the case of multi-component foods. High concentrations of DNA were isolated by this procedure, and their purity was also optimal. As for the remaining isolation procedures, they have proven to be less effective. In procedure no. 1, the problem may have been that the cell wall was not sufficiently disrupted and thus it was not possible to isolate DNA at higher concentrations. This shortcoming was addressed in procedure no. 2 and 3, when sterile zirconium and glass beads were added to the samples, which serve precisely to mechanically disrupt the cell wall (Table 3).

### 3.4. Comparison of Yield of Individual Isolation Procedures

A comparison of individual isolation procedures in terms of yield (Figure 2) shows that the most advantageous was procedure no. 3. By this procedure we managed to isolate on average DNA with a concentration of 89.6 ± 9.3 for single-component products and 77 ± 15.6 for multi-component products. The procedure was the most time consuming and required very thorough sample preparation before isolation. In insulation procedure no. 3, a commercial kit was used to isolate DNA from micro-mycetes. However, our results also indicate its suitability in isolating DNA from plant cells.

By isolation procedure no. 2, we obtained DNA at an average concentration of 50.1 ± 2.7 from single-component samples and 47.6 ± 11.7 from multi-component samples. We used a combination of zirconium and glass beads, Proteinase K, and Triton-X in this procedure. Zirconium and glass beads are used to mechanically disrupt the cell wall, and proteinase K to degrade proteins and to rapidly inactivate endogenous nucleases such as RNases and DNases. Triton-X serves to lyse cells and also to permeabilize the living cell membrane. However, this combination was not more effective than the use of the commercial Fungal DNA Mini Kit isolation procedure no. 3.

Procedure no. 1 proved to be the least efficient, with the help of which we managed to obtain DNA with a concentration of 28.2 ± 1.7 for single-component products and 21.4 ± 5.7 for multi-component products.

As can be seen from Table 1, Table 2 and Table 3, statistical analysis confirmed significant differences (*p* < 0.001) in the determined average DNA concentration between the individual isolation procedures in all product samples.

### 3.5. Detection of Gluten in Food Samples by the PCR Method

The PCR method is used in various areas of biological research and is considered to be the most significant discovery of the 1980s. It allows selective propagation of a desired stretch of DNA of up to a million copies in a test tube in vitro. For accurate and fast detection of gluten by PCR methods, high-quality and complete DNA in appropriate concentration and the highest possible purity is required in order to avoid false-negative or false-positive results. According to our findings, insulation procedure no. 3 was the most suitable. Therefore, for the subsequent PCR reaction, the isolated DNA obtained from the samples was used as a template by method no. 3.

The PCR reaction revealed the presence of gluten in guaranteed gluten-free products, but also in naturally gluten-free foods. These were mainly gluten-free sponge cakes, gluten-free “Brusienky” biscuits, cocoa powder, “3in1” coffee, and instant coffee.

As can be seen in Figure 3, according to the intensity of radiation of individual fragments under UV light, we can state the presence of larger amounts of gluten in samples of gluten-free sponge cakes and to a lesser extent in samples of gluten-free biscuits “Brusienky”, cocoa powder, coffee “3in1”, and instant coffee, although the initial concentration of isolated DNA samples to the PCR reaction was approximately the same. The presence of gluten in these samples can be explained by the contamination during their production or the presence of “hidden” gluten in the additives.

As the results show, the detection limit of the PCR method is lower, as gluten was also detected in foods that were declared gluten-free (i.e., had gluten less than 20 ppm). The method is more sensitive as it can detect the presence of gluten up to 0.16 ppm, with officially used methods for quantifying gluten in food ranging from 4 to 5 ppm gluten [18,28].

## 4. Discussion

Prior to the actual detection of gluten in foods by PCR, three different isolation procedures were used to obtain quality and pure DNA.

The first procedure was based on the addition of Triton-X. According to Borner et al. [29], it is one of the most widely used non-ionic surfactants for cell lysis for the extraction of proteins and other cell organelles or for the permeabilization of living cell membranes. However, if a large amount is added, or if the cells are exposed to long-term exposure to Triton-X, the cells will die. This toxicity of Triton molecules is due to the disruptive effect of its polar nature on the hydrogen bonds present in the cell lipid bilayer, leading to the destruction of the compactness and integrity of the lipid membrane. The incorporation of detergent monomer into the lipid membrane begins at low concentrations. This leads to disruption of the cell structure and possible excessive permeabilization of the cell membrane at concentrations above the critical micelle concentration from the bilayer micellar junction.

In the second procedure, we used sterile zircon and glass beads and Proteinase K, and after incubation we added Triton-X. Proteinase K is extremely effective on native proteins and is therefore used to rapidly inactivate endogenous nucleases such as RNases and DNases. Due to this property, proteinase K is particularly suitable for isolating native RNA and DNA from tissues or cell lines. This enzyme promotes cell lysis by activating bacterial autolytic factor. It is particularly suitable for isolating nucleic acids for amplification reactions. Proteinase K is also used to analyze membrane structures by modifying proteins and glycoproteins on cell surfaces [30].

The third procedure was to add zirconium and glass beads, Proteinase K and a lysis solution that was included in the commercial isolation kit. Based on the achieved results, we can evaluate that procedure no. 3 was the most advantageous from the point of view of gluten determination, because with this procedure we managed to detect gluten in samples of gluten-free sponge cakes and also in samples of gluten-free biscuits “Brusienky”, cocoa powder, coffee “3in1”, and instant coffee. Despite the label on the packaging that states ‘gluten-free food’, such cross-contamination can occur if food containing gluten is also processed in the common facility. Cross-contamination occurs when gluten particles are transferred from one object to another, which increases the risk of exposure and adverse effects to those who are unable to tolerate gluten [31].

Gluten is a protein found in wheat, rye, barley, triticale, and other cereals. To make it easier for consumers to follow a gluten-free diet, many manufacturers have introduced the production of a gluten-free version of many foods. In order for these foods to be called gluten-free, they must contain less than 20 mg/kg of gluten [31]. However, even consumers who try to consume only gluten-free foods can consume this gluten in the form of so-called hidden gluten. In this respect, it is necessary to know which foods and products can be contaminated in this way and ultimately negatively affect consumer health [32].

The first risky group of foods that may contain hidden gluten are marinades and sauces. Gluten can be found in sauces thickened with flour, or also in those that are flavored with malt vinegar or soy sauce, as these products are mainly made with the addition of wheat. In this case, it is advisable to choose gluten-free soy sauce or naturally gluten-free Japanese tamari sauce. In addition, it should be noted that wheat flour is traditionally mixed with butter, creating a backing that forms the basis of many sauces [33].

The second, and very common source of hidden gluten is processed meat and meat products, and imitation crab meat or various seafood. Sausages, meatloaf and minced meat often contain wheat-based fillers. Some semi-finished products, such as hamburger meat pancakes, can be supplemented with bread, which helps to bind the meat and improves its texture, which is why it is appropriate for celiacs to consume meat marked “100% minced meat” [33].

Another risk of gluten contamination comes from various alternatives to vegetarian meat. Many of these products are made with the addition of seitan, also known as wheat gluten or wheat meat. Others are made from gluten-containing flour or breadcrumbs, which serve as binders in food [34]. Other foods that can cause hidden gluten are drinks and alcohol. Gluten can be found in flavored coffees or teas. Many types of instant coffee contain gluten as a filler, and coffees flavored with powdered milk also contain wheat [35]. These claims are also confirmed by the results of this paper, where the presence of very small amounts of gluten was found in labeled or naturally gluten-free foods. As for alcohol, many species are made from gluten-containing grain. The distillation process should theoretically eliminate gluten proteins, but not all companies distill their products so thoroughly that complete removal occurs. Wine is a naturally gluten-free alcoholic drink [36]. Every person who reacts negatively to gluten intake tolerates different amounts in food. This depends in particular on the disease that the patient has been diagnosed with regarding this protein. If the individual does not tolerate even small amounts of gluten, this is so-called gluten allergy. A person can develop this disease at any time during their life, and they do not have to be born with it. It causes symptoms such as abdominal pain, cramps, bloating or diarrhea. After examining a small intestinal specimen, allergy sufferers do not provide a positive finding, but rather they have only positive specific blood antibodies, typical of gluten allergy. There is no autoimmune reaction in the body, nor does this cause inflammation such as celiac disease. While celiac disease is a lifelong disease, gluten allergy sufferers can return to a normal diet after a period of time. However, what is very important is that gluten allergy sufferers do not tolerate even a trace amount of gluten and therefore they must follow an even stricter gluten-free diet than celiacs [37].

## 5. Conclusions

The main contribution of this work was to propose a more efficient method for the isolation of DNA gluten from foods of plant origin. Polymerase chain reaction has been used to detect gluten at the molecular level in naturally gluten-free and gluten-free foods. The detection of gluten at the molecular level in gluten-free products points to a reassessment of the legally set limits on the concentration of gluten in foods, which we call gluten-free, as even trace amounts of gluten can cause health problems in sensitive patients. Monitoring the presence of gluten in foods is very important, especially in patients with gluten-related eating disorders. In these people, a gluten-free diet has been shown to be an effective treatment, but several studies report that up to 70% of celiacs do not have a fully recovered small intestine despite a strict diet. This is mainly due to unintentional consumption of hidden gluten in food.

## Figures and Tables

**Figure 1 foods-11-02011-f001:**
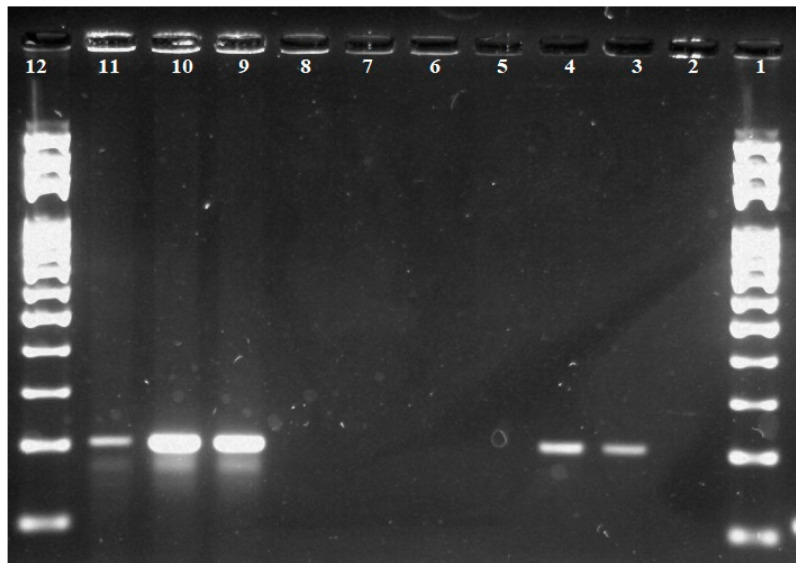
Comparison and verification of three isolation procedures for samples of wheat semi-coarse flour and dark wheat bread. Lane 1—100 bp standard; lane 2—negative control; lanes 3, 6, 9—semi-coarse wheat flour sample (PCR product = 200 bp); lanes 4, 7, 10—wheat bread sample (PCR product = 200 bp); lanes 5, 8, 11—sample of gluten-free sponge cake sample. Lanes 3 to 5—isolation procedure 2; lanes 6 to 8—isolation procedure 1; lanes 9 to 11—isolation procedure 3. Lane 12—100 bp standard.

**Figure 2 foods-11-02011-f002:**
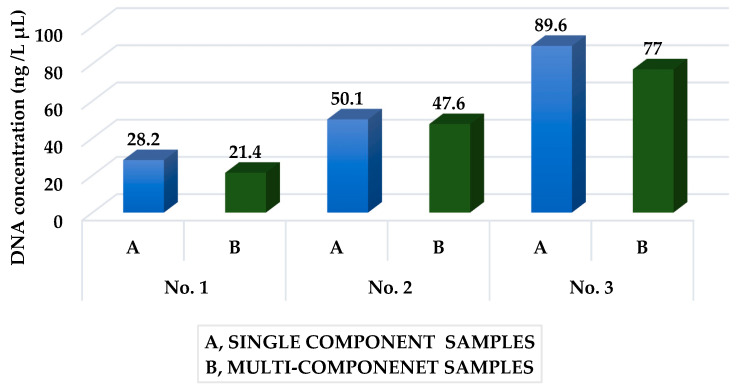
DNA concentration isolated by isolation procedure no. 1, 2 and 3.

**Figure 3 foods-11-02011-f003:**
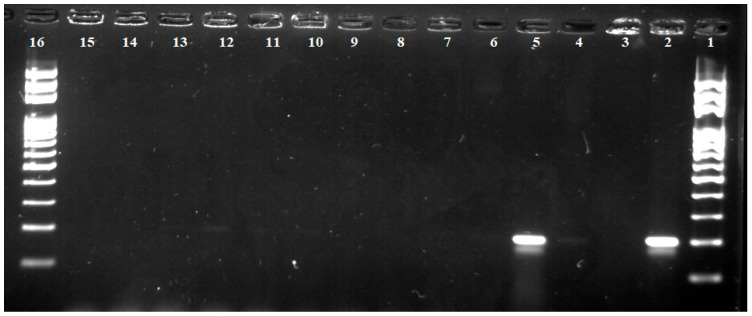
Detection of gluten in foods of plant origin (PCR product-200 bp). Lane 1—100 bp standard; lane 2—positive control (wheat flour); lane 3—negative control 1; lane 4—gluten-free “Brusienky”; lane 5—gluten-free sponge cakes; lane 6—“Crispins” corn sticks; lane 7—vanilla powder pudding; lane 8—gluten-free flour “Promix UNI”; lane 9—gluten-free pasta “Felicia Bio”; lane 10—corn crisps; lane 11—instant coffee; lane 12—cocoa powder; lane 13—coffee “3in1”; lane 14—corn starch; lane 15—negative control 2; Lane 16—100 bp standard.

**Table 1 foods-11-02011-t001:** Means and their standard deviations (SD) of the concentrations of isolated DNA samples from positive controls obtained by three isolation procedures.

Positive Control	Isolation Procedures						*p* Value
	No. 1		No. 2		No. 3		
	DNA Concentration (ng/µL)	DNA Purity	DNA Concentration (ng/µL)	DNA Purity	DNA Concentration (ng/µL)	DNA Purity	
Dark Wheat Bread	14.8 ± 0.6 ^c^	1.4 ± 0.0	50.0 ± 1.0 ^b^	1.7 ± 0.0	91.0 ± 1.0 ^a^	2.0 ± 0.0	<0.001
Semi-Coarse Wheat Flour	23.0 ± 2.7 ^c^	1.2 ± 0.0	69.0 ± 1.0 ^b^	1.6 ± 0.0	99.7 ± 0.6 ^a^	1.8 ± 0.0	<0.001

^a, b, c^—Means within a row different superscript differ (*p* < 0.05).

**Table 2 foods-11-02011-t002:** Means and their standard deviations (SD) of the concentrations of isolated DNA samples from positive controls obtained by three isolation procedures.

Products Samples	Isolation Procedures						*p* Value
	No. 1		No. 2		No. 3		
	DNA Concentration (ng/µL)	DNA Purity	DNA Concentration (ng/µL)	DNA Purity	DNA Concentration (ng/µL)	DNA Purity	
Cocoa Powder	24.3 ± 0.6 ^c^	1.0 ± 0.0	56.3 ± 0.6 ^b^	1.7 ± 0.0	86.0 ± 0.0 ^a^	2.0 ± 0.0	<0.001
Corn Starch	28.4 ± 0.7 ^c^	1.5 ± 0.0	50.0 ± 1.0 ^b^	1.7 ± 0.0	90.3 ± 0.6 ^a^	2.1 ± 0.0	<0.001
Vanilla Pudding Powder	29.7 ± 0.6 ^c^	1.4 ± 0.0	51.6 ± 0.6 ^b^	1.5 ± 0.0	99.0 ± 0.0 ^a^	2.2 ± 0.0	<0.001
Instant Coffee	28.6 ± 1.5^c^	1.4 ± 0.0	47.0 ± 1.0 ^b^	1.8 ± 0.0	80.4 ± 0.7 ^a^	1.8 ± 0.0	<0.001
Gluten-Free Flour “Promix UNI”	26.3 ± 1.2 ^c^	1.0 ± 0.0	51.6 ± 0.6 ^b^	1.6 ± 0.0	89.3 ± 1.2 ^a^	1.9 ± 0.0	<0.001

^a, b, c^—Means within a row different superscript differ (*p* < 0.05).

**Table 3 foods-11-02011-t003:** Means and their standard deviations (SD) of isolated DNA from multi-component.

Products Samples	Isolation Procedures						*p* Value
	No. 1		No. 2		No. 3		
	DNA Concentration (ng/µL)	DNA Purity	DNA Concentration (ng/µL)	DNA Purity	DNA Concentration (ng/µL)	DNA Purity	
Gluten-Free Sponge Cakes	19.3 ± 1.5 ^c^	1.3 ± 0.0	46.3 ± 1.5 ^b^	1.5 ± 0.0	88.0 ± 1.0 ^a^	1.9 ± 0.0	<0.001
Corn Crisps	23.3 ± 1.5 ^c^	1.4 ± 0.0	48.7 ± 0.6 ^b^	1.4 ± 0.0	78.0 ± 0.0 ^a^	2.1 ± 0.0	<0.001
Gluten-Free “Brusienky”	19.0 ± 2.0 ^c^	0.9 ± 0.0	47.0 ± 1.0 ^b^	1.0 ± 0.0	68.4 ± 0.7 ^a^	1.9 ± 0.0	<0.001
Corn Sticks “Crispins”	21.0 ± 1.7 ^c^	1.4 ± 0.0	49.0 ± 0.0 ^b^	1.4 ± 0.0	87.3 ± 0.6 ^a^	2.2 ± 0.0	<0.001
Gluten-Free Pasta “Felicia Bio”	16.0 ± 1.7 ^c^	1.4 ± 0.0	35.3 ± 0.6 ^b^	1.8 ± 0.0	59.0 ± 1.0 ^a^	1.8 ± 0.0	<0.001
Coffee “3in1”.	27.3 ± 0.6 ^c^	1.0 ± 0.0	58.4 ± 0.5 ^b^	1.6 ± 0.0	84.6 ± 0.5 ^a^	1.8 ± 0.0	<0.001

Food samples obtained by three isolation procedures. ^a, b, c^—Means within a row in different superscript differ (*p* < 0.05).

## Data Availability

The data presented in this study are available upon request from the corresponding author.

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
