# Peer review of "Detection of Gluten in Gluten-Free Foods of Plant Origin"

_foods, 2022, doi:10.3390/foods11142011_

Round 1
Reviewer 1 Report
I am very grateful you for the invitation to review the manuscript foods-1775094 by Výrostková and co-authors " Detection of gluten in gluten-free foods of plant origin". The work deals with the issue of gluten detection in gluten-free products of plant origin. The work is interesting but needs adjustments to increase the quality of the material.
Comments:
- The authors should make it clear that the main problem studied is the standardization of a more accurate methodology. The analysis of the presence of gluten takes a back seat, mainly due to low sampling. In this case, the authors need to clarify in the introduction (problems associated with the analysis of gluten in plant products);
- Page 1, Abstract: Briefly include information about the presence of gluten in products.
- Page 1, Abstract: Include numerical information about % gluten detection in the evaluated products, to demonstrate the magnitude of the problem more easily.
- Page 1, Line 21: Change the repeated keywords by different words from the title to expand the search system.
- Page 1, Line 25-27: Better explain the technological role of structure in bakery products.
- Page 1, Line 41-42: Specify the permission of gluten in gluten-free products around the world by legislation.
- Standardize the use of units throughout the text: “30 minutes” or “5 min”, for example.
- Page 4, Lines 164-168: The information has already been presented before. Avoid unnecessary repetitions. In addition, part of the information should be used in the introduction to justify the study (need for a methodology to qualify the presence of gluten).
- Page 4, Line 193: Explain the reason for greater purity in multi-component foods.
- Page 8, Lines 295-304: The information has already been presented in the previous items. Authors should check the possibility of withdrawing or condensing to avoid unnecessary repetition.
- Specify whether, in the case of the evaluated products, the legislation allows the presence of low concentrations to be called gluten-free, as detection would be possible even when in low concentration.
- Page 9, Lines 360-361: The authors need to clarify whether the gluten quantification is carried out and whether it is below what the legislation recommends to be considered gluten-free. Detection is not to be confused with being above parameters.
- Page 10, Lines 378-380: The greatest finding is related to the determination of the best methodology since the quantity and sampling of products are very small.
Author Responses
Dear Reviewer
Please find attached our revised research article „ Detection of gluten in gluten-free foods of plant origin” written by Jana Výrostková, Ivana Regecová*, František Zigo, Slavomír Marcinčák, Ivona Kožárová, Mariana Kováčová and Daniela Bertová which we would like to submit for consideration to the Foods in special issue " Sustainable Food Systems and Food Policy for Healthy Diets ".
We would like to thank the reviewer for comments making this manuscript clearer and more reliable. All recommendations have been accepted by the authors. Thank you for considering this manuscript.
Yours Sincerely,
Ivana Regecová, DVM, PhD.
Corresponding Author

Reviewer 2 Report
foods-1775094-peer-review-v1
Detection of gluten in gluten-free foods of plant origin
The manuscript under evaluation successfully compares three methods for isolating gluten DNA from foods of plant origin and subsequent detection by PCR.
Some suggestions are given below
1- Section Introduction Lines 29-38, page 1
Please include any additional citations specific to these lines.
2- Section Introduction
Please include a paragraph about the official method used in your country for the quantification of gluten in foods, advantages and disadvantages of the method, what standards the official method meets. Justify the need to develop new or more efficient methods. Mention the detection limits of the official method at the trace level. Please include a link or specific bibliography on the official methods in your country.
As an example I suggest a link to a South American country (Argentina) http://www.anmat.gov.ar/enfermedad_celiaca/Guia_determinacion_gluten_en_alimentos.pdf
3- Section 2. Materials and Methods
Samples
Please indicate the country of origin of the purchased samples. Be more specific or comment on the samples, on their labeling, they say gluten-free or specify traces.
4- Section 2.1. DNA isolation
Please include bibliographic references, if possible, that support each of the methods used.
5- In the different methodologies developed, was any Certified Reference Material (CRM) used?.
6- At the end of the results section, please include a paragraph that mentions the detection limits of the three methods used and compares them with the official method.
7- After the changes made in the manuscript, please review and modify the conclusions if necessary.
Author Response
Dear Reviewer
Please find attached our revised research article „ Detection of gluten in gluten-free foods of plant origin” written by Jana Výrostková, Ivana Regecová*, František Zigo, Slavomír Marcinčák, Ivona Kožárová, Mariana Kováčová and Daniela Bertová which we would like to submit for consideration to the Foods in special issue " Sustainable Food Systems and Food Policy for Healthy Diets ".
We would like to thank the reviewer for comments making this manuscript clearer and more reliable. All recommendations have been accepted by the authors. Thank you for considering this manuscript.
Yours Sincerely,
Ivana Regecová, DVM, PhD.
Corresponding Author

Round 2
Reviewer 2 Report
The authors have responded to all suggestions made.